# The Global Prader–Willi Syndrome Registry: Development, Launch, and Early Demographics

**DOI:** 10.3390/genes10090713

**Published:** 2019-09-14

**Authors:** Jessica Bohonowych, Jennifer Miller, Shawn E. McCandless, Theresa V. Strong

**Affiliations:** 1Foundation for Prader–Willi Research, Walnut, CA 91789, USA; jessica.bohonowych@fpwr.org; 2Department of Pediatrics, University of Florida School of Medicine, Gainesville, FL 32611, USA; millejl@peds.ufl.edu; 3Section of Genetics and Metabolism, Department of Pediatrics, University of Colorado School of Medicine and Children’s Hospital Colorado, Aurora, CO, USA

**Keywords:** Prader–Willi syndrome, registry, natural history

## Abstract

Advances in technologies offer new opportunities to collect and integrate data from a broad range of sources to advance the understanding of rare diseases and support the development of new treatments. Prader–Willi syndrome (PWS) is a rare, complex neurodevelopmental disorder, which has a variable and incompletely understood natural history. PWS is characterized by early failure to thrive, followed by the onset of excessive appetite (hyperphagia). Additional characteristics include multiple endocrine abnormalities, hypotonia, hypogonadism, sleep disturbances, a challenging neurobehavioral phenotype, and cognitive disability. The Foundation for Prader–Willi Research’s Global PWS Registry is one of more than twenty-five registries developed to date through the National Organization of Rare Disorders (NORD) IAMRARE Registry Program. The Registry consists of surveys covering general medical history, system-specific clinical complications, diet, medication and supplement use, as well as behavior, mental health, and social information. Information is primarily parent/caregiver entered. The platform is flexible and allows addition of new surveys, including updatable and longitudinal surveys. Launched in 2015, the PWS Registry has enrolled 1696 participants from 37 countries, with 23,550 surveys completed. This resource can improve the understanding of PWS natural history and support medical product development for PWS.

## 1. Introduction

Prader–Willi syndrome (PWS) is a rare genetic disorder with a birth incidence of 1/10,000 to 1/30,000, and an estimated prevalence of approximately 10,000 to 20,000 living individuals in the United States [1,2,3]. It affects males and females equally, as well as all races and ethnicities [3]. PWS is caused by the absence of paternally-expressed, imprinted genes on chromosome 15q11–13. Loss of activity can occur by one of three major genetic mechanisms: Microdeletion of the paternally inherited chromosome (60%–70% of cases, further divided into type I and type II, 7 megabases (Mb) and 5 Mb in size, respectively); maternal uniparental disomy (UPD) subsequent to trisomic rescue (30%–40% of cases); or mutation/epimutation of the Prader–Willi syndrome/Angelman syndrome (PWS/AS) imprinting center (~3% of cases) [1,2,3]. Consensus clinical diagnostic criteria for PWS were reported in 1993 [4], but the phenotype is variable, and a definitive molecular diagnostic test for PWS is available, which accurately detects >99% of cases [3]. The PWS region of chromosome 15 is unusually complex and includes protein coding genes as well as long noncoding RNAs and embedded, short noncoding RNAs [5]. The exact contribution of each of the genes in the region to the overall phenotype has not been definitely determined and, to date, the molecular mechanisms underlying the PWS phenotype are not fully understood. However, PWS is thought to be primarily a disorder of hypothalamic function, impacting multiple systems throughout the body [1,3,6].

PWS is associated with a constellation of symptoms that significantly negatively impact quality of life for affected individuals and their families. The initial clinical course of PWS is characterized by hypotonia in infants, with decreased movement, lethargy, feeding difficulties, and failure to thrive. A defining feature of PWS is the change in appetite over time, with the onset of hyperphagia (an unrelenting, pathologically excessive appetite) sometime after early childhood [3,7]. Whereas infants with PWS do not show normal signs of hunger and often require feeding via nasogastric tubes or other assistive means, feeding improves in young children and subsequently progresses to hyperphagia during childhood. Adolescents and adults with PWS will become morbidly obese if strict environmental controls restricting food intake are not implemented. Additional abnormalities associated with PWS include growth hormone deficiency, hypogonadotropic hypogonadism, sleep disorders, reduced pain sensitivity, poor bone health, decreased gastrointestinal motility, and scoliosis. Aspects such as central adrenal insufficiency, seizures, hypothyroidism, and hypoglycemia occur at frequencies higher than the normal population, but are not present in all individuals [1,3,6].

In addition to somatic symptoms, intellectual disability (ID) and neuropsychiatric issues are present to some degree in all individuals with PWS [3,8,9]. Individuals with PWS typically exhibit a characteristic behavioral phenotype that includes cognitive rigidity, heightened anxiety, severe temper outbursts, obsessive-compulsive behaviors, and self-injurious behaviors. Adolescents and adults are at risk of mental illness and autistic symptomatology is common, particularly in those with PWS by UPD. Hyperphagia-driven behaviors include food-seeking behavior, hoarding or foraging for food, eating of inappropriate food items (e.g., raw or discarded food), stealing food or money to buy food, and intense psychological stress and behavioral disturbances associated with food denial. Behavioral issues and an inability to control food intake represent the major impediments to independent living for individuals with PWS, and opportunities for community engagement, employment, independent living, and social activities are highly constrained by these issues.

There is a wide range of phenotypic variability in PWS, some of which is not attributable to the different genetic subtypes. With respect to major classes of genetic subtypes (deletion vs. UPD vs. imprinting mutations), overall there is more phenotypic variability within subtypes than across subtypes, including for features such as hyperphagia, scoliosis, sleep disruption, pain insensitivity, growth hormone deficiency and other endocrine dysfunctions, self-injurious behavior (skin picking), and obesity. Notable exceptions are autism spectrum disorder and psychosis, both of which are significantly more common in the UPD subtype compared to deletion, while major depressive disorder is more common in the deletion subtype [10,11].

Current treatments for PWS are limited and, to date, focus on the treatment of endocrine abnormalities with hormone replacement therapy [12,13]. Growth hormone (GH) is FDA-approved for treating children with PWS and is increasingly prescribed in infants and adults [14]. GH is effective in normalizing growth and improving body composition in PWS but has no effect on hyperphagia. To date, no FDA-approved drugs have proven effective in controlling appetite and food-related behavior in PWS. Thus, parent education, restricted diet, and environmental control are the only options for avoiding morbid obesity in PWS. However, a number of potential drugs and devices are currently in preclinical and clinical development and may provide new therapeutic options for those with PWS in coming years [15,16,17,18,19,20,21,22,23]. While the number of therapeutic avenues being pursued is encouraging, a better understanding of natural history, as well as a robust database of potential participants for clinical studies, can facilitate the development and evaluation of novel therapeutics for PWS.

The US National Institutes of Health (NIH)-funded PWS Rare Disease Consortium studied approximately 350 individuals with PWS over an eight-year period (2006–2014) [24], providing valuable insights into the natural history of PWS. However, many critical gaps remain. Long-term longitudinal description of disease manifestations, onset and progression of characteristic behaviors, as well the aging process in PWS individuals are incompletely defined. Gaps also exist in understanding the co-morbidities of PWS; the relationship between genotype and phenotype; the impact of genetic variants and environmental considerations on the clinical course of disease; and treatment outcomes across a broad population of individuals with PWS. Finally, as care changes and novel therapies become available, a flexible platform for gathering data on current real-world experiences and outcomes in the PWS population is needed.

The Global Prader–Willi Syndrome Registry (www.pwsregistry.org) was launched in 2015 by the Foundation for Prader–Willi Research (FPWR) in collaboration with leading clinicians to accelerate research and support the development of treatments for PWS. The Registry aims to document the natural history of PWS; understand the full spectrum of PWS features across the entire population; identify unmet medical needs, rare complications, and understudied areas; facilitate partnerships with stakeholders; expedite completion of clinical trials; guide the development of standards of care; and allow participants to centrally store their PWS medical data.

## 2. Materials and Methods

Registry Architecture and Security: The Global PWS Registry is web-based and is hosted on the National Organization for Rare Disorders (NORD) “IAMRARE” Registry Platform. NORD’s Registry platform was developed with input from experts at the NIH, patient advocacy groups, and the US Food and Drug Administration (FDA). The Global PWS Registry is compliant with US Health Information Privacy Laws, FDA regulations on electronic records, and the security requirements of the European Union General Data Protection Regulation (GDPR). Registry data is only accessed by registry study personnel. De-identified data can be shared with researchers and other stakeholders as per the Registry protocol. All protocols, surveys, and recruitment materials are reviewed and approved by a central Institutional Review Board (IRB).

Registry Governance: The Global PWS Registry is governed by the Global PWS Registry Advisory Board. The members of the Advisory Board represent stakeholders in the PWS and rare disease communities including PWS patient groups, parents, caregivers, clinicians, and experts in Registry development and governance. The Advisory Board is tasked with (1) reviewing requests for de-identified data by researchers, clinicians, and other stakeholders; (2) reviewing requests from sponsors to send out IRB-approved recruitment materials to participants that meet study inclusion/exclusion criteria; and (3) reviewing proposals to run research projects through the Registry, particularly those that may include development of new surveys.

Survey Development: Survey development was a collaborative effort, led by the Registry investigators. The development team included input from various stakeholders with expertise in PWS including clinicians, academic researchers, parents, and other caregivers. The process was informed by resources including publications on best practices in Registry development [25], communications with other disease registries, and recommendations from the NORD IAMRARE Team. Surveys were beta tested with a small cohort of PWS families that provided feedback on any questions where the language was confusing or too technical. Beta testers also provided input on improving or expanding answer options for questions, which was incorporated into the final surveys.

Data Curation: Registry administrators continually review new participant accounts and remove duplicate accounts, or accounts created in error as determined by duplicate names and/or email addresses. Accounts with invalid contact information are also removed. On an ongoing basis, data points within the registry are curated for date of birth, diagnosis, and relationship of the respondent or person managing the account, to the person with PWS. Any errors that are identified are corrected by Registry administrators. Curation is done through communicating with respondents directly when their contact preferences grant permission, and also through comparing and validating common data elements across surveys within the Registry. An example of this includes comparing the date of birth on the account, with the date of birth on an uploaded medical record within the account. Data for this publication was accessed on 30 May 2019, and includes all completed responses for each question. Some questions were not completed by all respondents, thus the number of responses per question varies.

## 3. Results

### 3.1. Registry Design and Initial Recruitment

The Global PWS Registry is open to parents, guardians, caregivers, and to individuals with PWS. More than 90% of Registry accounts are managed by parents, guardians, or legal authorized representatives. However, there is a subset of accounts that are managed by self-reporting individuals with PWS.

The Global PWS Registry launched in April 2015 with 35 surveys (Table 1). The surveys are designed to capture the full spectrum and natural history of PWS, including birth history and diagnosis, a comprehensive battery of specific medical systems, as well as demographics, well-being, and quality of life. The initial 35 surveys are all updatable and serve as a comprehensive record over time, wherein participants return on an annual basis to add new information. In addition to these updatable surveys, the Registry platform also supports longitudinal surveys which repeatedly capture the same information at time point intervals.

### 3.2. Enrollment and Survey Completion

The Global PWS Registry currently has 1696 participants who have completed the consent process (Table 2). Survey completion rates vary, with 21% of participants having completed all 35 of the initial surveys. Thirty-five percent (35%) of participants have completed between five to 34 surveys, and 17% have completed one to five surveys. There is also a group of 27% of participants that have consented, but not yet submitted any surveys. This all translates to a total of 23,550 survey submissions (Table 1). The “Getting Started” survey, which is the first survey of the Registry, has the most submissions at 1207. The surveys with the least number of submissions include the six surveys related to medications and supplements, as well as the two surveys covering “Behavior” and “Hippotherapy, Psychotherapy, and Behavioral Therapies”.

### 3.3. Geographical Distribution of Registry Participants

In total, 37 countries are represented within the Global PWS Registry (Table 3). Participants are predominantly from the United States and U.S. Territories, making up over 75% of Registry participants. Canada has the second highest representation at 8.9%. Australia, the United Kingdom, and New Zealand each make up between 1% to 3% of the Registry. The remaining countries each constitute less than 1% of the Registry.

Within the United States, there are Registry participants from all 50 states, as well as from the District of Columbia, and Puerto Rico (Table 4). The four states with the highest percentage of Registry participants are California (7.6%), Texas (6.8%), New York (5.7%), and Florida (5.1%). There are eight states that each have between 3% to 5% of US Registry participants, 19 states each with 1% to 3%, and the remaining states, districts, and territories each have less than 1%. Within Canada, eight of the 10 provinces are represented with over 40% of Canadian participants residing in Ontario (Table 5). The next highest percentages are British Columbia (17.4%), Alberta (15.1%), and Quebec (14%).

### 3.4. Race, Ethnicity, Age and Gender Demographics

Of those reporting race and ethnicity, the majority of Registry participants (person with PWS) are Caucasian (85.1%) and non-Hispanic or Latino (82%) (Table 6 and Table 7). The age of participants at the time of enrollment is predominantly under the age of 15 years old (68%) (Figure 1). The two- to five-year old and six- to 10-year old age groups makeup the highest percentages, at 18% and 20%, respectively. These percentages demonstrate that the majority of participants within the Global PWS Registry are minors, and that there is an underrepresentation of young adults and adults with PWS. As expected, the gender distribution of participants is relatively equal at 51% male and 49% female (Figure 2).

Participants within the Registry represent 35 countries throughout the world. Participants from the United States make up the majority of the Registry. Shown are the responses from those specifying their country of origin; 627 participants did not answer this question.

### 3.5. PWS Genetic Subtype

Of participants within the Registry who have a reported genetic subtype, 49.3% reported having PWS due to a paternal deletion and 35.6% reported having uniparental disomy (UPD) (Figure 3). Approximately 3.3% of participants reported having an imprinting defect. A small percentage of Registry respondents (1.7%) indicated that the participant fell into an “other” category. Open ended responses to a follow up question for “other” showed that these individuals include participants with PWS due translocations, atypical microdeletions, or “acquired PWS” (e.g., PWS-like symptoms subsequent to a hypothalamic tumor). In addition, a small number of participants (n = 2) in the Global PWS Registry have a diagnosis of Schaaf-Yang syndrome [26], a PWS-related disorder caused by truncating mutations in the *MAGEL2* gene, which resides in the PWS-region of chromosome 15. Genetic subtype is unknown for 10.1% of participants. These participants may have been diagnosed with PWS based on a clinical diagnosis only (more common in older participants), or they may have received a molecular diagnosis of PWS based on DNA methylation analysis only, without additional testing to discern genetic subtype.

Of those reporting genetic subtype, maternal age at conception was also reported for 593 participants (352 PWS with PWS due to deletion and 241 with PWS due to UPD). The mean maternal age for PWS participants with the UPD genetic subtype was significantly older than for those with PWS due to deletion (34.8 years, SD 5.6 for UPD compared to 29.2 years, SD 5.3 for deletion; *p* < 0.0001), consistent with advanced maternal age and trisomy 15 rescue as the genetic mechanism underlying the UPD subtype [1].

### 3.6. Community Engagement

The initial recruitment for the Registry involved a multipronged approach including social media, newsletters, and e-mails in cooperation with PWS advocacy groups, presentations at PWS family conferences and fundraisers, brochures at PWS clinics, and informational webinars. A number of helpful resources have been created to guide families through the registration and informed about the consent process. These include a “getting started video”, a “getting started PDF”, a private Facebook group, webinars, brochures, FAQ documents, and direct engagement at family conferences. As the Registry has developed, additional outreach to PWS group homes, clinics, physicians, and international patient groups has occurred. An important aspect of Registry communication is the return of results to the PWS community. Through the Registry platform, participants are immediately able to view graphs of de-identified aggregate data for any surveys they have completed. In addition, at regular intervals, infographics addressing aspects of PWS are developed from Registry data and shared with the community through social media and email (Figure 4).

## 4. Discussion

Since it was launched in 2015, the Global PWS Registry has steadily grown in enrollment and now includes an estimated 5% to 10% of the US PWS population, with individuals from 37 countries represented in all. The IAMRARE platform provides the flexibility of both updatable as well as longitudinal surveys. These formats provide a means to capture the medical history of onset and severity of PWS symptoms throughout a participant’s life, as well as to track changes over shorter periods of time for aspects of the PWS phenotype such as behavior, anxiety, or sleep. NORD continues to make improvements and enhancements to the IAMRARE platform to support survey development, data analysis, and the development of sub-studies focusing on specific groups or phenotypes. Additional long-term goals include integrating other data sources, including real-world data. The richness of the data collected to date within the PWS Registry is already supporting the development of novel therapeutics for PWS and facilitating partnerships with academic clinicians and pharmaceutical industry stakeholders. However, several challenges remain, most notably recruiting a population to the Registry that better reflects the demographics and entire spectrum of PWS (age, nationality, race and ethnicity, socio-economic status, PWS experience).

Although age estimates for the general PWS population are difficult to establish, the age distribution currently represented in the Registry is likely younger than the actual PWS population. Mortality rates are higher in PWS than the general population across all ages [27]. With only 13% of current Registry participants over the age of 25, there is a large population of adults that is not represented in the Registry. Challenges for enrolling and completing surveys for adults with PWS include navigating guardianship/consent for adults not living at home and/or for adults whose parents are deceased, assembling a long-accumulated history of medical records, lack of knowledge about the database, and engaging an older parent population that may be less comfortable with a web-based format. However, this adult PWS population group represents a critical segment of the Registry in gaining an understanding of the natural history of PWS. Moreover, there are early reports of premature aging in the PWS population [28]. This is a poorly understood area and the Registry can help identify health issues in adults with PWS, as well as direct priorities for clinical care in adults with PWS.

The Registry has enrolled a significant portion of the US PWS population, as well as participants from numerous other countries. However, more than 87% of the Registry is from the US and Canada, and additional work is needed to establish a participant population that more accurately reflects the global PWS population. The main hurdle in addressing this challenge is that the Registry is currently only available in English. There are plans to eventually offer the Registry in additional languages. This will entail medically-certified translations of all of the questions and informed consents. It will also require software enhancements to the IAMRARE platform to provide the site in multiple languages and to support appropriate data coding so that answers from multiple language versions of the same questionnaire can be queried together. Although challenging, the ability to offer the Registry in multiple languages will facilitate comparison of health care, treatment standards, and quality of life across countries.

With respect to population profile, currently, minorities make up a very small percentage of Registry participants. For example, 1.5% and 3% of Registry participants identify as African American and Asian, respectively. According to the 2010 US Census, those minority populations comprise 12.6% and 4.8% of US residents. Considering that the incidence of PWS is not associated with race or ethnicity, and that US participants comprise almost 80% of the Registry, these percentages highlight the underrepresentation of non-whites within the Global PWS Registry and suggest that the current Registry population is not fully representative of the ethnic and racial diversity of the expected PWS population. Offering the registry in multiple languages, as described above, may allow a more diverse US population to enter the Registry as well. Developing additional recruitment strategies to improve representation that more accurately reflects the racial and ethnic diversity in the US is also a priority. Collaborating with patient groups, clinics, and group homes to understand how to better reach these underrepresented demographic groups, as well as any hurdles/challenges to their participation in the Registry, is critical. Working with these stakeholders will also help broaden the phenotypic spectrum of individuals with PWS (i.e., individuals who are doing very well, as well as those who are more severely impacted). All of these demographics are critical to understanding the full spectrum of the disorder.

Outside of improving demographic representation within the Registry, additional concerns around this research tool include the potential limitations of parent/caregiver-reported data, and the challenge of completing all surveys for each participant with annual updates. Although the clinical accuracy of parent/caregiver-reported data may have limitations, the IAMRARE platform allows clinical notes and records to be uploaded to support or supplement the parent/caregiver-reported information. Encouraging parents/caregivers to upload diagnostic reports to allow reported genetic subtype to be confirmed, for example, is a current priority. In addition, engaging with the PWS community and demonstrating the value of the Registry is critical to increasing survey completion and having participants regularly update their profiles.

Return of results to the community is a critical component of retaining active participation in the Registry. Infographics highlighting data from topics of interest are shared through newsletters and social media, and the input of the community on topics of interest is encouraged. Data has also been presented at several family conferences. Launching new surveys, and special projects or sub-studies with researchers, also brings people back into the Registry.

With growing participation, the Global PWS Registry is poised to continue leveraging de-identified data through collaborations with researchers, industry, and other partners to advance the understanding of PWS, direct efforts in basic and clinical research, and support the development of novel therapies for this challenging disorder.

## Figures and Tables

**Figure 1 genes-10-00713-f001:**
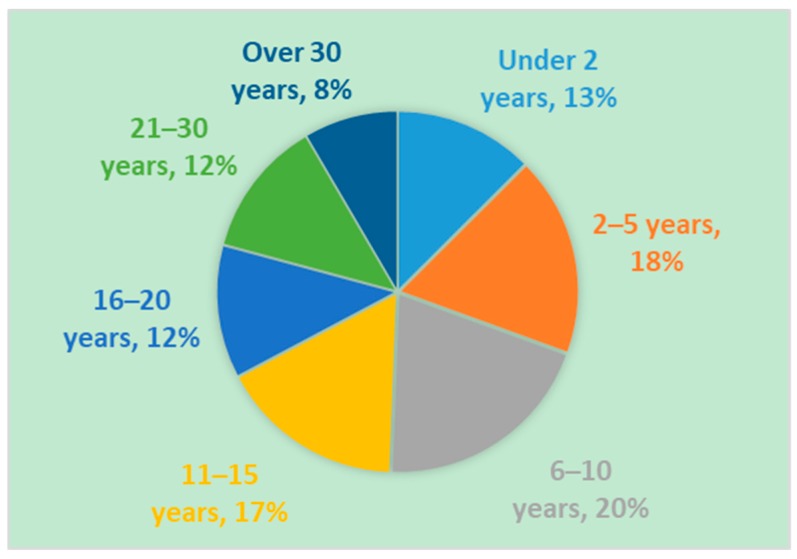
Age of Registry participants at time of enrollment. Registry participants range in age from newborn infants to adults. The majority of registry participants are under 15 years of age.

**Figure 2 genes-10-00713-f002:**
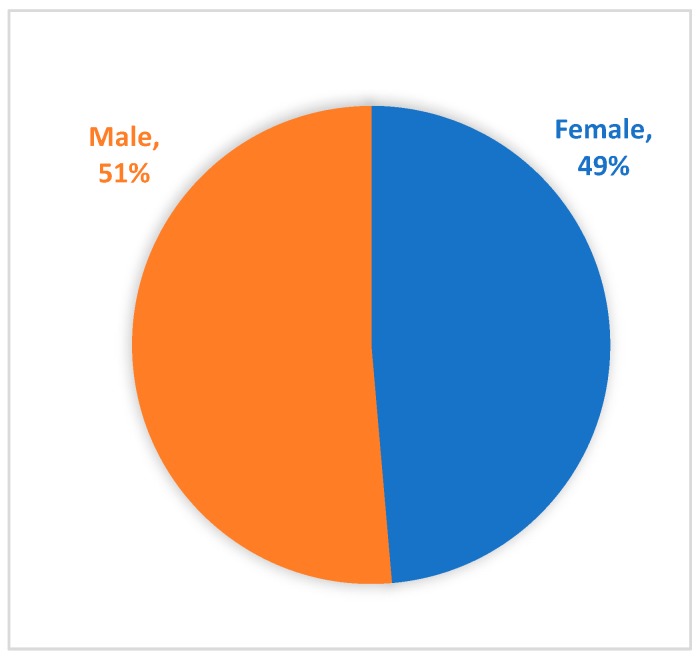
Gender of Registry participants.

**Figure 3 genes-10-00713-f003:**
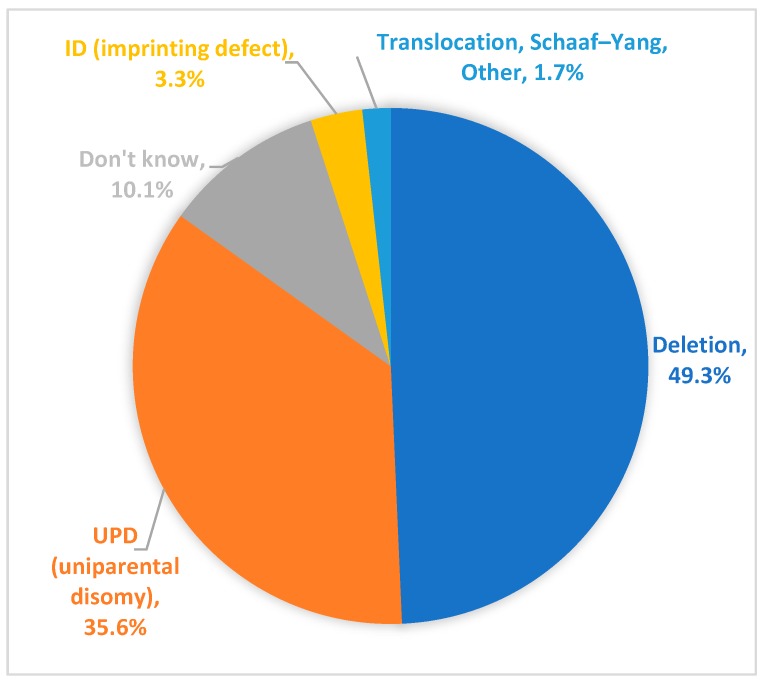
PWS genetic subtype of Registry participants. PWS by deletion is the predominant genetic subtype of participants within the Registry. PWS by uniparental disomy (UPD) is the second most common diagnosis. PWS by imprinting defect, translocation, and micro-deletions are also represented within the Registry. Approximately 10% of Registry participants do not know their PWS genetic subtype.

**Figure 4 genes-10-00713-f004:**
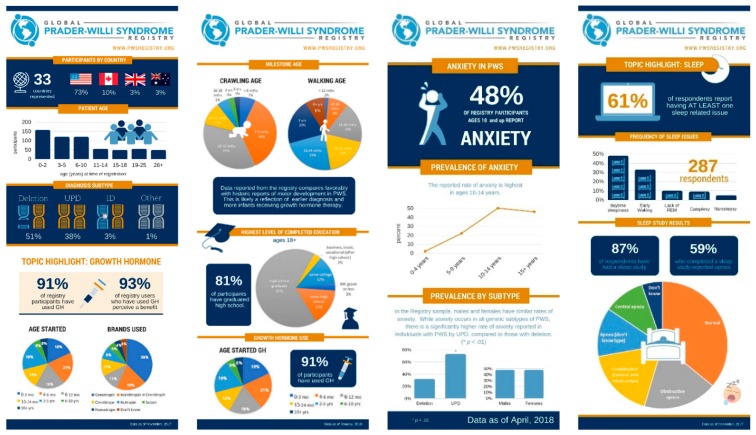
Infographics of Registry data shared with the PWS community. De-identified aggregate data is exported from the Registry, analyzed, and represented in infographics, and shared through social media, print newsletters, and other communications. Each infographic is designed to highlight a specific topic area of interest to PWS families, clinicians, and other stakeholders within the PWS community. Infographics and accompanying descriptions are available at https://www.fpwr.org/blog.

**Table 1 genes-10-00713-t001:** Surveys included in the Global Prader–Willi syndrome (PWS) Registry launch, and the number of submissions per survey. The initial launch of the PWS Registry included 35 surveys. Surveys have been completed by varying numbers of participants.

Survey Name	# of Participants
Getting Started	1207
Contact Information	1074
Participant Demographics	969
Research Trials	959
Diagnosis	952
Pregnancy History	861
Birth History	821
Biological Family History	810
General Medical History	787
Neurological History	767
Vision History	756
Developmental Milestones	708
Psychological and Mental Health	682
Speech, Occupational, and Physical Therapy	654
Sleep History	613
Pulmonary History	609
Gastrointestinal History	605
Sexual and Reproductive History	603
Dental History	584
Education and Employment	582
Endocrinological History	573
Dermatological History	570
Orthopedic History	565
Ear, Nose, and Throat Health History	560
Sociodemographic and Socioeconomic	555
Well–Being	548
Nutritional Phase and Diet	547
Medications–Endocrinology	545
Supplements A–M	526
Medications–Cardiology, GI, and Others	520
Supplements N-Z and Others	511
Medications–Psychiatric A–M	509
Medications–Psychiatric N–Z and Others	502
Hippotherapy, Psycotherapy, and Behavioral Therapies	488
Behavior	428
**TOTAL**	**23,550**

**Table 2 genes-10-00713-t002:** Survey completion percentages. The number of surveys completed by each participant ranges from 0 surveys to all 35 surveys.

# of Surveys Completed	# of Participants	% of Participants
0 surveys	463	27%
1–5 surveys	280	17%
6–10 surveys	208	12%
11–20 surveys	179	11%
21–34 surveys	209	12%
All surveys	357	21%
**Total**	**1696**	**100%**

**Table 3 genes-10-00713-t003:** Distribution of participants within the Global PWS Registry by country.

Country	# of Participants	% of Participants
United States	842	78.8%
Canada	90	8.4%
Australia	29	2.7%
United Kingdom	22	2.1%
New Zealand	12	1.1%
Mexico	7	0.7%
Spain	7	0.7%
France	5	0.5%
Germany	4	0.4%
Ireland	4	0.4%
South Africa	4	0.4%
Belgium	3	0.3%
Brazil	3	0.3%
India	3	0.3%
United States Minor Outlying Islands	3	0.3%
Austria	2	0.2%
China	2	0.2%
Colombia	2	0.2%
Finland	2	0.2%
Greece	2	0.2%
Israel	2	0.2%
Sweden	2	0.2%
Bangladesh	1	0.1%
Bermuda	1	0.1%
Bulgaria	1	0.1%
Croatia	1	0.1%
Denmark	1	0.1%
Hong Kong	1	0.1%
Italy	1	0.1%
Lebanon	1	0.1%
Malaysia	1	0.1%
Netherlands	1	0.1%
Norway	1	0.1%
Poland	1	0.1%
Portugal	1	0.1%
Puerto Rico	1	0.1%
Singapore	1	0.1%
Taiwan, Province Of China	1	0.1%
United Arab Emirates	1	0.1%
**TOTAL**	**1069**	**100%**

**Table 4 genes-10-00713-t004:** Distribution of US participants within the Global PWS Registry by state. Participants within the Registry represent all 50 states within the United States. Of participants that live in the United States, California, Texas, New York, and Florida have the highest reported number of participants.

US State	# of Participants	% of US Participants
Alabama	14	1.7%
Alaska	7	0.8%
Arizona	22	2.7%
Arkansas	6	0.7%
California	63	7.6%
Colorado	19	2.3%
Connecticut	14	1.7%
Delaware	1	0.1%
District of Columbia	1	0.1%
Florida	42	5.1%
Georgia	31	3.7%
Hawaii	4	0.5%
Idaho	4	0.5%
Illinois	31	3.7%
Indiana	22	2.7%
Iowa	9	1.1%
Kansas	12	1.5%
Kentucky	7	0.8%
Louisiana	9	1.1%
Maine	3	0.4%
Maryland	11	1.3%
Massachusetts	24	2.9%
Michigan	27	3.3%
Minnesota	19	2.3%
Mississippi	3	0.4%
Missouri	17	2.1%
Montana	5	0.6%
Nebraska	7	0.8%
Nevada	4	0.5%
New Hampshire	4	0.5%
New Jersey	23	2.8%
New Mexico	7	0.8%
New York	47	5.7%
North Carolina	25	3.0%
North Dakota	1	0.1%
Ohio	39	4.7%
Oklahoma	7	0.8%
Oregon	8	1.0%
Pennsylvania	34	4.1%
Puerto Rico	1	0.1%
Rhode Island	5	0.6%
South Carolina	9	1.1%
South Dakota	2	0.2%
Tennessee	17	2.1%
Texas	56	6.8%
Utah	17	2.1%
Vermont	1	0.1%
Virginia	23	2.8%
Washington	30	3.6%
West Virginia	4	0.5%
Wisconsin	29	3.5%
Wyoming	0	0.0%
**TOTAL**	**827**	**100%**

**Table 5 genes-10-00713-t005:** Distribution of Canadian participants within the Global PWS Registry by province. Participants within the Registry represent eight of the Canadian provinces. Participants from Ontario make up the largest group of Canadian participants. Shown are the responses from those specifying a province; four Canadian participants did not answer this question.

Canadian Province	# of Participants	% of Participants
Alberta	13	15.1%
British Columbia	15	17.4%
Manitoba	2	2.3%
New Brunswick	2	2.3%
Newfoundland & Labrador	1	1.2%
Ontario	37	43.0%
Quebec	12	14.0%
Saskatchewan	4	4.7%
**TOTAL**	**86**	**100%**

**Table 6 genes-10-00713-t006:** Race of participants within the Global PWS Registry.

Race	# of Participants	% of Participants
Caucasian	796	85.1%
Multi–Ethnic	66	7.1%
Asian	28	3.0%
Black or African American	14	1.5%
Other	17	1.8%
Native Hawaiian or Other Pacific Islander	3	0.3%
Prefer not to Answer	6	0.6%
American Indian or Alaska Native	4	0.4%
Don’t know	1	0.1%
**TOTAL**	**935**	**100%**

**Table 7 genes-10-00713-t007:** Ethnicity of participants within the Global PWS Registry.

Ethnicity	# of Participants	% of Participants
Non–Hispanic or Latino	629	82.0%
Hispanic or Latino	73	9.5%
Ashkenazi Jewish	19	2.5%
Unknown	30	3.9%
Prefer not to answer	16	2.1%
**TOTAL**	**767**	**100%**

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
