# Peer review of "The Global Prader–Willi Syndrome Registry: Development, Launch, and Early Demographics"

_genes, 2019, doi:10.3390/genes10090713_

Round 1

Reviewer 1 Report

This is a well written, interesting and clear manuscript. I'm only slightly confused by some of the totals in some of the tables, and suggest you verify them - or I might have misinterpreted what was being shown.

There are very minor comments, with text that needs addressing highlighted in yellow and a comment box to the right.

I wish we had more such databases to draw on!!  I found myself wondering about co-morbidities and how they're captured, which will be interesting to look into at some point.

Well done!

Author Response

We thank the Reviewer for the careful review and helpful comments/edits. 

A concern was raised by discrepancies in the total number of responses, which is not consistent across all Tables. This is due to the fact that not every participant responded to every question (for example, 90 participants indicated they are residents of Canada, but only 86 answered the follow up question about which Province they reside in).  We have attempted to clarify this in the text (added a last sentence to Methods: Some questions were not completed by all respondents, thus the number of responses per question varies.) and in the Table legends (for example, additional sentence in Table 3 legend: Shown are the responses from those specifying their country of origin; 627 participants did not answer this question.)   

Additional minor editing suggestions from the Reviewer have been incorporated into the revised text. 

Reviewer 2 Report

The authors report current status of the Global Prader-Willi Syndrome Registry. Prader-Willi syndrome (PWS) is a rare neurodevelopmental disorder that is caused by genetic and epigenetic defects in imprinting genes on chromosome 15q. Although accurate molecular diagnosis for PWS has already established, disease phenotypes and natural courses are substantially heterogeneous across patients. To date, there is no FDA-approved drugs for controlling appetite and food-related behavior in PWS. The registry was launched in 2015 to conduct long-term longitudinal descriptions on a wide spectrum of clinical courses of the disease and treatment outcomes from a large number of patients across many populations. The manuscript concisely explains how the registry has been constructed. A total of 1,696 patients have been recruited from 37 countries. The authors briefry summarize characteristics of the participants and achievement rates of surveys. The authors developed nice infographics to help users to visually understand progress of registry data. The efforts of the authors to recruit a large number of PWS individuals, conduct a set of surveys, and summarize the data will be of great help to understand clinical courses of PWS and to develop new therapeutic treatments including drugs and social cares.   

My questions are:

A substantial number of the participants (one fourth) could not complete any surveys. Is there any possibilities that individuals whose conditions are severe could not complete surveys?

Infographics are well designed. Does the registry plan to open the infographics to publicly accessible website and so on? If so, please provide the information about the address.

Minor Comment:

Page 6 Line 212: “able” was duplicated.

Author Response

We thank the reviewer for the thoughtful review and helpful comments.

With respect to the concern regarding individuals with severe disease not being able to complete the surveys, the vast majority of respondents who complete the surveys are parents answering the surveys on behalf of their child with PWS. This is true even for the adults with PWS, since intellectual disability is limiting for most individuals with PWS.  Thus low survey completion rates are not likely to be directly related to more severe disease.  However, it is possible that parent/caregivers who have more medically complex children may be less likely to complete Registry surveys given the demands of care. Our recruitment presentations to patient families emphasize the need to capture the entire spectrum of PWS severity, and encourages survey completion by all participants. 

Regarding public accessibility to infographics - we have shared the infographics with the patient and medical community through email newsletters, conference presentations, social media and blogs.  The text of Figure 4 has been modified to include the web address of the Foundation for Prader-Willi Research's blog, which includes entries about the PWS Registry infographics.